# Improvement of Network Flow Using Multi-Commodity Flow Problem

**Takato Fukugami \***  **and Tomofumi Matsuzawa** 

Department of Information Sciences, Tokyo University of Science, Yamazaki, Chiba 278-8510, Japan; t-matsu@is.noda.tus.ac.jp
\* Correspondence: fukugami.tus@gmail.com

**Abstract:** In recent years, Internet traffic has increased due to its widespread use. This can be attributed to the growth of social games on smartphones and video distribution services with increasingly high image quality. In these situations, a routing mechanism is required to control congestion, but most existing routing protocols select a single optimal path. This causes the load to be concentrated on certain links, increasing the risk of congestion. In addition to the optimal path, the network has redundant paths leading to the destination node. In this study, we propose a multipath control with multi-commodity flow problem. Comparing the proposed method with OSPF, which is single-path control, and OSPF-ECMP, which is multipath control, we confirmed that the proposed method records higher packet arrival rates. This is expected to reduce congestion.

**Keywords:** multi-commodity flow problem; routing; load balancing; OpenFlow



## 1. Introduction

In recent years, network traffic has been increasing with the increase in applications using the Internet. This can be attributed to the growth of social games and video distribution services, which are becoming higher quality. Traffic volume per subscription is expected to increase approximately 14 times by 2030 compared to current traffic volumes [1]. In addition to PCs and smartphones, the development of the Internet of Things (IoT) [2] has led to an era in which automobiles and home appliances are connected to the Internet, and various terminals are sending packets. By 2030, the number of IoT devices is expected to exceed about 29 billion [3]. Routing must be performed to ensure that these services are available even under conditions of packet growth.

Currently, most routing protocols select a single optimal path, such as open shortest path first (OSPF) [4]. OSPF is a link-state routing protocol that belongs to the interior gateway protocol (IGP) and uses a cost set for each link. Dijkstra's algorithm [5] is used for route calculation, and the path that minimizes the total cost to the destination node is taken as the optimal path. If there are multiple equal-cost paths, OSPF can distinguish between them, but it arbitrarily uses one path by default. OSPF is effective when the optimal path has a large bandwidth. However, when the optimal path has a narrow bandwidth and a large amount of traffic flows over it, the risk of congestion is high and the desired performance is not achieved. In such cases, the current OSPF is problematic.

In addition to the optimal path, redundant paths lead to the network's destination node. Therefore, load balancing using multiple paths is effective in reducing congestion [6]. Load balancing using multiple equal-cost paths is referred to as equal-cost multipath (ECMP) [7]. A round-robin divides traffic equally among all equal-cost paths. However, its performance depends on the number of equal-cost paths. The paths are limited, and other redundant paths can be used in the network as a whole.

In this study, we propose a routing method that uses multiple paths with the multi-commodity flow problem [8] based on the maximum flow problem [9]. The commodity in the multi-commodity flow problem is the combination of source and destination. No

simplified augmenting path algorithm is currently known. Linear programming [5] and a full polynomial time approximation scheme [10] are used as solution methods. This improves the throughput of the entire network and suppresses congestion. For this purpose, we implement the necessary functions on OpenFlow [11] and evaluate its performance.

## 2. Previous Research

Many modified algorithms based on Dijkstra's algorithm have been proposed in studies considering shortest paths. Kadry et al. reduced the computational complexity by reducing the number of iterations [12]. Wei et al. improved the algorithm so that the maximum load path can be found [13]. However, these proposals do not achieve load balancing.

Previous studies have attempted to achieve load balancing. HiQoS [14] computes multiple paths for all pairs using Dijkstra's algorithm for early recovery from link loss. However, this is not designed for large traffic flows. The distributed flow-by-flow fair routing (DFFR) algorithm [15] routes so that each switch has an equal load on all equal-cost paths. DFFR does this without rerouting or splitting flows, thus preserving TCP performance. However, DFFR assumes that all links are homogeneous and is not optimal in network topologies where this is not the case. Sorted-GFF [16] uses a fixed threshold set for the bandwidth of the link and recalculates the path if that threshold is exceeded. However, a fixed threshold cannot handle variability, such as differences in device performance. Moreover, when large traffic flows exceed the threshold value, the path is recalculated frequently, which places a heavy burden on the controller. When the threshold is increased, recalculation can be suppressed, but the load on the link cannot be detected. Therefore, a large amount of traffic cannot be assumed to flow. The proposed method uses a multi-product flow problem to compute paths using the entire network, which improves performance and allows the system to handle large amounts of traffic.

## 3. Preliminary Experiment

Table 1 shows the multi-commodity flow problem formulated as a linear programming problem when there are 1 and $k$ commodities. The point set is $V$, the edge set is $E$, the directed graph consisting of $V$ and $E$ is $G = (V, E)$, the non-negative capacity function defined on the edges is $u : E \rightarrow R^+$, any edge is $e \in E$, the starting point is $s \in V$ and the end point is $t \in V$, the flow on each edge is $x : E \rightarrow R^+$.

**Table 1.** Formulation of the multi-commodity flow problem.

| | **1 Commodity** | **$k$ Commodities** |
|---|---|---|
| Objective Function (Max) | $\sum_{e \in \delta^+(s)} x_e - \sum_{e \in \delta^-(s)} x_e$ | $\sum_{n=1}^{k} \left( \sum_{e \in \delta^+(s^n)} x_e - \sum_{e \in \delta^-(s^n)} x_e \right)$ |
| Capacity Constraint | $0 \le x_e \le u_e$ | $0 \le x_e^n, \quad \sum_{n=1}^{k} x_e^n \le u_e$ |
| Flow Conservation Law | $\sum_{e \in \delta^+(v)} x_e - \sum_{e \in \delta^-(v)} x_e = 0$ | $\sum_{e \in \delta^+(v)} x_e^n - \sum_{e \in \delta^-(v)} x_e^n = 0$ |

The above equation maximizes the total flow of all commodities. Here, it is expected that some cases will occur where the flow of one commodity is large and that of another commodity is small. We confirm this hypothesis by conducting experiments using the GNU Linear Programming Kit (GLPK) [17].

The version of GLPK is 5.0, and the solution method used for the linear programming problem is the simplex method [5].

Figure 1 shows the topology used in the experiment. There are two commodities in total, with commodity 1 flowing from node 1 to node 4, and commodity 2 flowing from node 2 to node 4. The maximum flow of this graph is 28 when the maximum flow problem is calculated only for commodity 1.

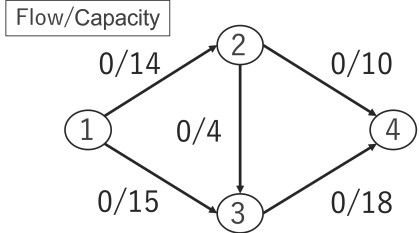

**Figure 1.** Pre-experiment topology.

Figure 2 shows the output of GLPK and its graphing as a result of a preliminary experiment.

```
No. Column name  St  Activity
------ ------------ -- -------------
    1 x[1,1,1]     NL            0
    2 x[1,1,2]     NL            0
    3 x[1,2,1]     B            10
    4 x[1,2,2]     NL            0
    5 x[1,3,1]     B            15
    6 x[1,3,2]     NL            0
    7 x[1,4,1]     B             0
    8 x[1,4,2]     NL            0
    9 x[2,1,1]     NL            0
   10 x[2,1,2]     B             0
   11 x[2,2,1]     NL            0
   12 x[2,2,2]     NL            0
   13 x[2,3,1]     NL            0
   14 x[2,3,2]     B             3
   15 x[2,4,1]     B            10
   16 x[2,4,2]     NL            0
   17 x[3,1,1]     NL            0
   18 x[3,1,2]     B             0
   19 x[3,2,1]     NL            0
   20 x[3,2,2]     NL            0
   21 x[3,3,1]     NL            0
   22 x[3,3,2]     NL            0
   23 x[3,4,1]     B            15
   24 x[3,4,2]     B             3
   25 x[4,1,1]     NL            0
   26 x[4,1,2]     NL            0
   27 x[4,2,1]     NL            0
   28 x[4,2,2]     NL            0
   29 x[4,3,1]     NL            0
   30 x[4,3,2]     NL            0
   31 x[4,4,1]     NL            0
   32 x[4,4,2]     NL            0
   33 totalflow[1] B            25
   34 totalflow[2] B             3
```

(**a**)

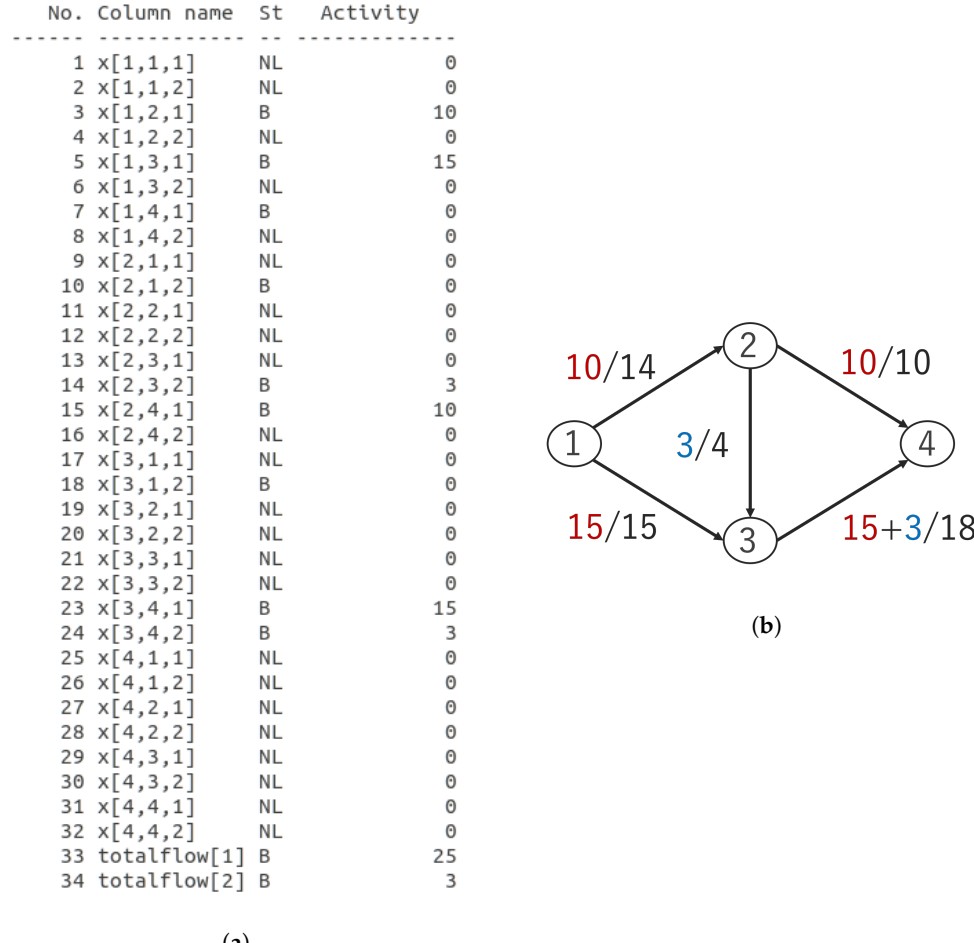

(**b**)

**Figure 2.** Result of preliminary experiment. (**a**) Result of GLPK execution. (**b**) Graph of the result of GLPK executions.

The first argument of $x$ in Figure 2a represents the starting point, the second argument represents the end point, and the third argument represents the commodity. The Activity column is the amount of each flow. The *total flow* in the last line is the total flow of each commodity, and the result of 28 was obtained as the maximum flow for the entire graph. However, commodity 1 inhibited the flow of commodity 2, resulting in a significant bias. As a commodity with a small flow is considered a single path, load balancing by multiple paths cannot be achieved. In addition, if the result of zero flow for one commodity is obtained, routing for that commodity cannot be performed because there is no path.

## 4. Proposed Method

### 4.1. Objective Function Using the Maximin Principle

Assuming total $k$ commodities exist, the constraint equation and objective function applying the maximin principle [18] are shown below.

**constraint equation:**

$$\sum_{e \in \delta^+(s^n)} x_e - \sum_{e \in \delta^-(s^n)} x_e \geq v \quad (n = 1, \ldots, k) \tag{1}$$

**objective function:**

$$\text{Max. } v \tag{2}$$

Equation (2) is intended to maximize the minimum flow among all commodities, not the maximum flow of each commodity. To increase the flow in a state where the flow of each commodity is fair, the objective function in Table 1 and the new weights $w_1$ and $w_2$ are combined into Equation (3). The $w_1$ and $w_2$ values are manually changed by the administrator according to the actual network conditions.

$$\text{Max. } w_1 v + w_2 \sum_{n=1}^{k} \left( \sum_{e \in \delta^+(s^n)} x_e - \sum_{e \in \delta^-(s^n)} x_e \right) \quad (w_1 + w_2 = 1) \tag{3}$$

### 4.2. Routing Using Multi-Commodity Flow Problem

We propose a routing method that performs load balancing with multiple paths using the multi-commodity flow problem. This improves the throughput of the entire network. The protocol used is OpenFlow.

The multi-commodity flow problem requires capacity on each edge. In OSPF, the cost that the router $m$ uses for the link $(m, n)$ with neighboring router $n$ can be manually set, but often follows Equation (4).

$$\text{cost}(m, n) = \text{ceil} \left( \frac{\text{reference bandwidth(bps)}}{\text{link bandwidth(bps)}} \right) \tag{4}$$

Equation (4) is inversely proportional to the per unit time performance of the link. By making the capacity inversely proportional to the value of the OSPF cost, it can reflect the performance of the link.

In addition, because it is necessary to know all the commodities to calculate the multi-commodity flow problem, the commodities are fixed from the beginning and the flow table is created by proactive control. For linear programming, we apply the method described in Section 4.1.

The source and destination pairs and paths are registered for each commodity in the flow table of the OpenFlow switch. If there are multiple paths, a select-type group table is used. Select-type selects one of the ports defined as a group in a round-robin fashion. The allocation ratio parameter sets the amount of flow obtained from the multi-commodity flow problem. Table 2 shows an example of the flow table and group table when 10 flows flow into an OpenFlow switch, 3 flows flow out from port 1, and 7 flows flow out from port 2.

**Table 2.** Example of switch.

| (a) Flow table | |
|---|---|
| **Match Field** | **Instruction** |
| (source, destination) | group 1 |

| (b) Group table | | | |
|---|---|---|---|
| **ID** | **Type** | **Parameter** | **Action** |
| 1 | select | 3<br>7 | output 1<br>output 2 |

## 5. Experiment

### 5.1. Objective Function Using the Maximin Principle

The experimental environment and topology are the same as in the preliminary experiment (Figure 1), and the output of GLPK and its graphical representation are shown in Figure 3. The value of weights $w_1$ and $w_2$ is 0.5. $v$, in the last line of Figure 3a, is the minimum flow among all commodities.

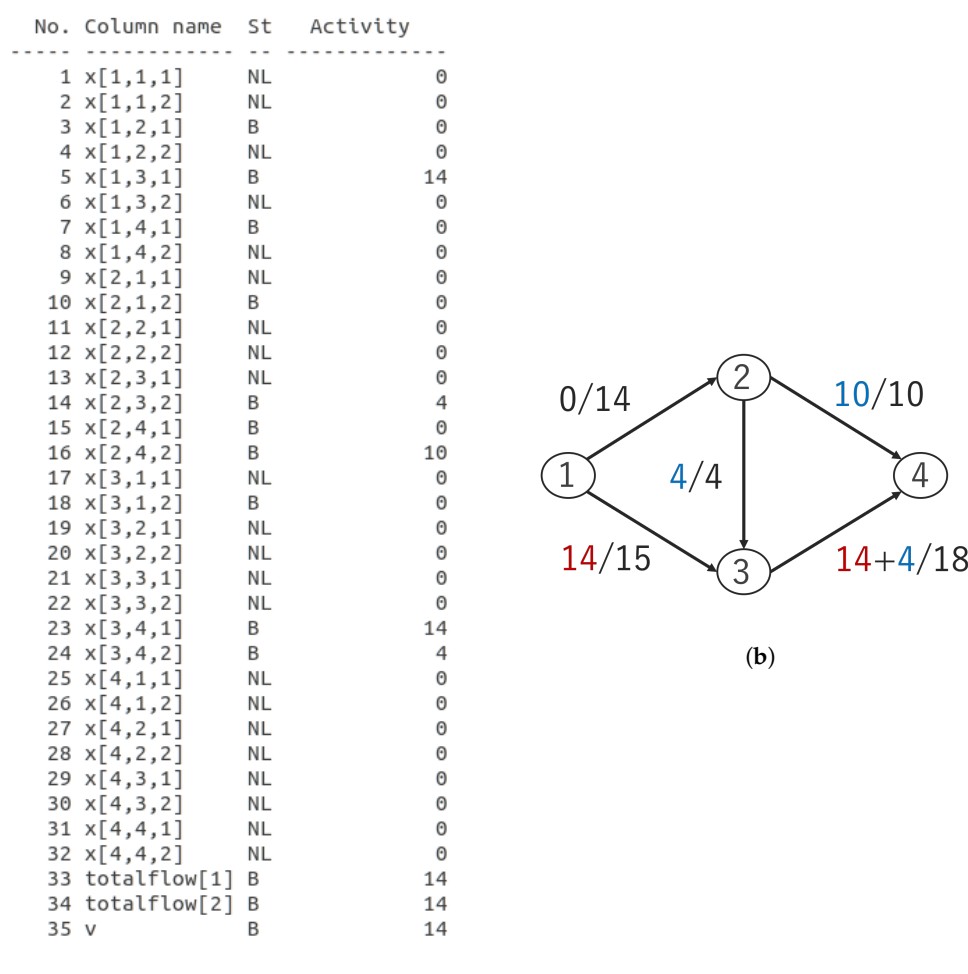

```
No. Column name  St   Activity
--- ------------ --  ------------
  1 x[1,1,1]     NL            0
  2 x[1,1,2]     NL            0
  3 x[1,2,1]     B             0
  4 x[1,2,2]     NL            0
  5 x[1,3,1]     B            14
  6 x[1,3,2]     NL            0
  7 x[1,4,1]     B             0
  8 x[1,4,2]     NL            0
  9 x[2,1,1]     NL            0
 10 x[2,1,2]     B             0
 11 x[2,2,1]     NL            0
 12 x[2,2,2]     NL            0
 13 x[2,3,1]     NL            0
 14 x[2,3,2]     B             4
 15 x[2,4,1]     B             0
 16 x[2,4,2]     B            10
 17 x[3,1,1]     NL            0
 18 x[3,1,2]     B             0
 19 x[3,2,1]     NL            0
 20 x[3,2,2]     NL            0
 21 x[3,3,1]     NL            0
 22 x[3,3,2]     NL            0
 23 x[3,4,1]     B            14
 24 x[3,4,2]     B             4
 25 x[4,1,1]     NL            0
 26 x[4,1,2]     NL            0
 27 x[4,2,1]     NL            0
 28 x[4,2,2]     NL            0
 29 x[4,3,1]     NL            0
 30 x[4,3,2]     NL            0
 31 x[4,4,1]     NL            0
 32 x[4,4,2]     NL            0
 33 totalflow[1] B            14
 34 totalflow[2] B            14
 35 v            B            14
```

(a)

**Figure 3.** Calculation results with the same topology as in the preliminary experiment. (**a**) Results of GLPK execution. (**b**) Graph of the result of GLPK executions.

The total flow of the two commodities was 28. The proportion of commodity 2 flows was larger than in the preliminary experiment (Figure 3).

Next, Figure 4 shows the result of the usual calculation and the result of the proposed method in a topology with three nodes in series. There are three commodities in total, with

commodity 1 flowing from node 1 to node 3, commodity 2 flowing from node 1 to node 2, and commodity 3 flowing from node 2 to node 3.

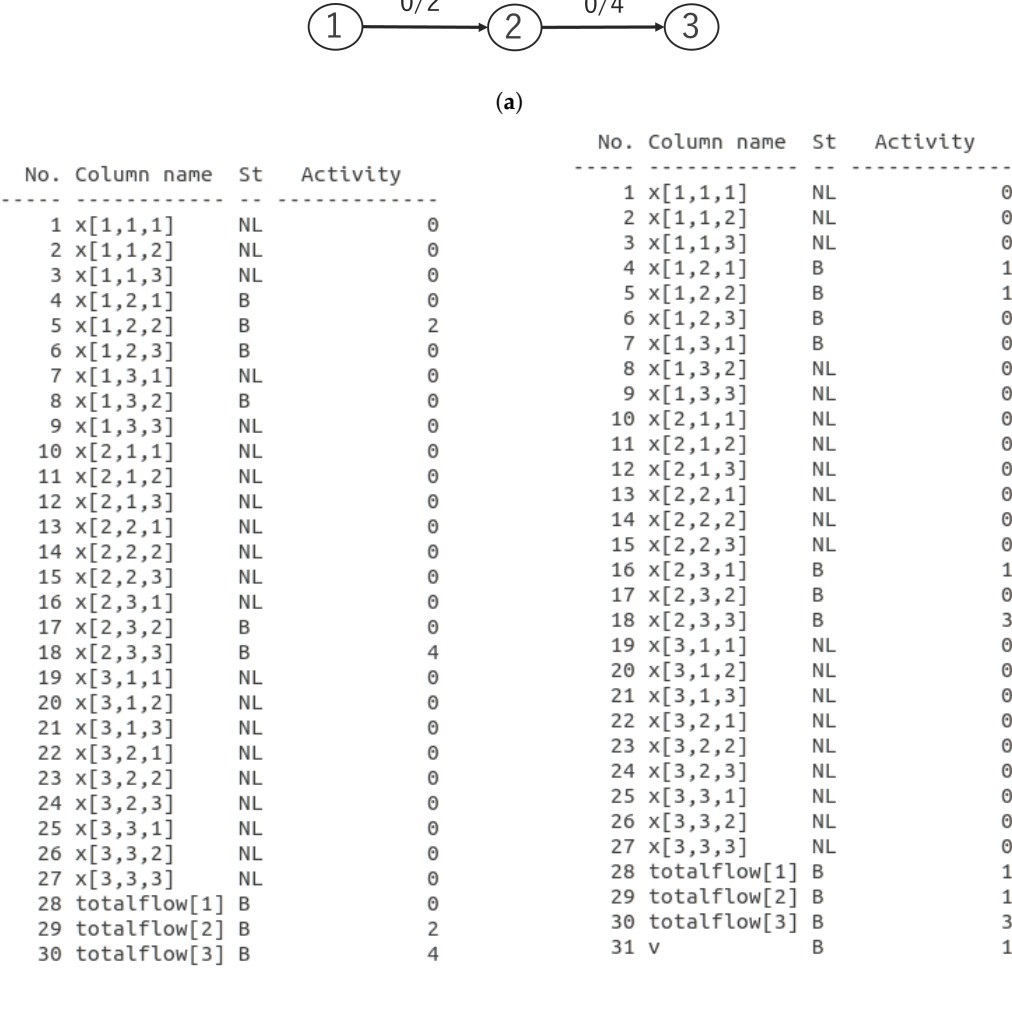

**Figure 4.** Calculation results in serial topology. (**a**) Serial topology. (**b**) Normal calculation result. (**c**) Calculation result using the proposed method.

The maximum flow was 6, but the flow of commodity 1 was 0 (Figure 4b). As the proposed method in Section 4.2 uses the result of the multi-commodity flow problem in the flow table and group table; the packets of commodity 1 do not flow. However, owing to the proposed method, each commodity had 1 or more flows, but the total flow of the three commodities was 5 (Figure 4c). This does not necessarily mean that the result of the calculation using the proposed method will be the maximum flow for the entire graph.

### 5.2. Routing Using Multi-Commodity Flow Problem

To confirm the superiority of the proposed method, we compared its performance against OSPF and OSPF-ECMP by generating multiple CBR (constant bit rate) UDP traffic to verify the packet arrival rate to the destination node and the average end-to-end delay. Experiment 1 compared the proposed method with OSPF, and Experiment 2 compared it with OSPF-ECMP.

Experiments were performed on version 3.29 of ns-3 [19]. The operating system run was Linux and the distribution was Ubuntu 18.04.6. OFSWITCH13 [20], provided as open source, was used to implement OpenFlow. This library provides switches and controllers that support OpenFlow 1.3. OSPF is not implemented in ns-3. Therefore, we created a

new class that inherits from *Ipv4RoutingProtocol*, which manages the routing protocol in ns-3, and created a routing table using Dijkstra's algorithm. In addition, We extended the OSPF class so that OSPF-ECMP selects one of the equal-cost paths in a round-robin fashion. The proposed method first solves a linear programming problem with the commodities set in the simulation. When the handshake is successful between the OpenFlow controller and the switch, the results are registered in each flow table by flow-mod or group-mod messages. We used the same topology for all experiments and added OpenFlow switch functionality to each node only in the implementation of the proposed method.

The experiment involved use of the Cost239 topology (Figure 5), which is modeled after European cities.

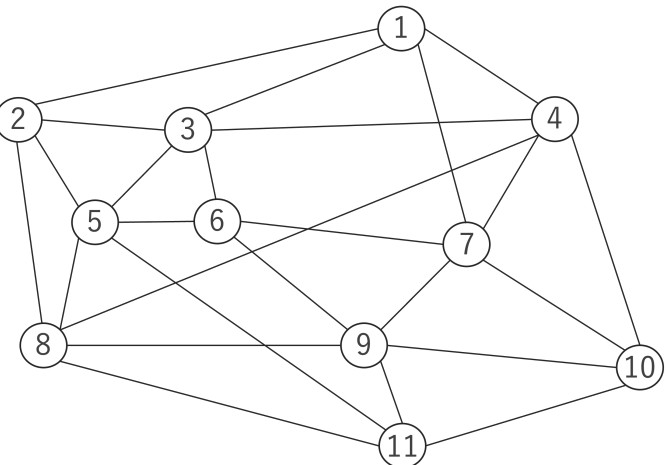

**Figure 5.** Cost239 topology.

Table 3 shows the parameters used in the experiment.

**Table 3.** Parameters of experiment.

| Parameter | Value |
|---|---|
| Packet Size | 1500 byte |
| Packet Transmission Time | 10 s |
| OSPF Reference Bandwidth | 10 Mbps |
| Link Bandwidth (Experiment 1) | Select a bandwidth with a uniform random number that results in a cost of 10 to 20. |
| Link Bandwidth (Experiment 2) | 1 Mbps |
| Bitrate (Experiment 1) | 100 kbps, 200 kbps, 300 kbps, 400 kbps, 500 kbps, 600 kbps, 700 kbps, 800 kbps, 900 kbps, 1 Mbps, 1.1 Mbps, 1.2 Mbps, 1.3 Mbps, 1.4 Mbps, 1.5 Mbps, 1.6 Mbps, 1.7 Mbps |
| Bitrate (Experiment 2) | 800 kbps, 900 kbps, 1 Mbps, 1.1 Mbps, 1.2 Mbps, 1.3 Mbps, 1.4 Mbps, 1.5 Mbps, 1.6 Mbps |

### 5.2.1. Results of Experiment 1

**Number of sources: 3, number of destinations: 1**

The destination for all three commodities is node 9, with node 1 as the source for commodity 1, node 2 as the source for commodity 2, and node 8 as the source for commodity 3. Before comparing with OSPF, the results of experiments with GLPK using the original objective function in Table 1 are shown in Figure 6. The results of verifying packet arrival rate with ns-3 are shown in Figure 7.

```
364 totalflow[1] B          55
365 totalflow[2] B          17
366 totalflow[3] B          10
```

**Figure 6.** Result of GLPK execution.

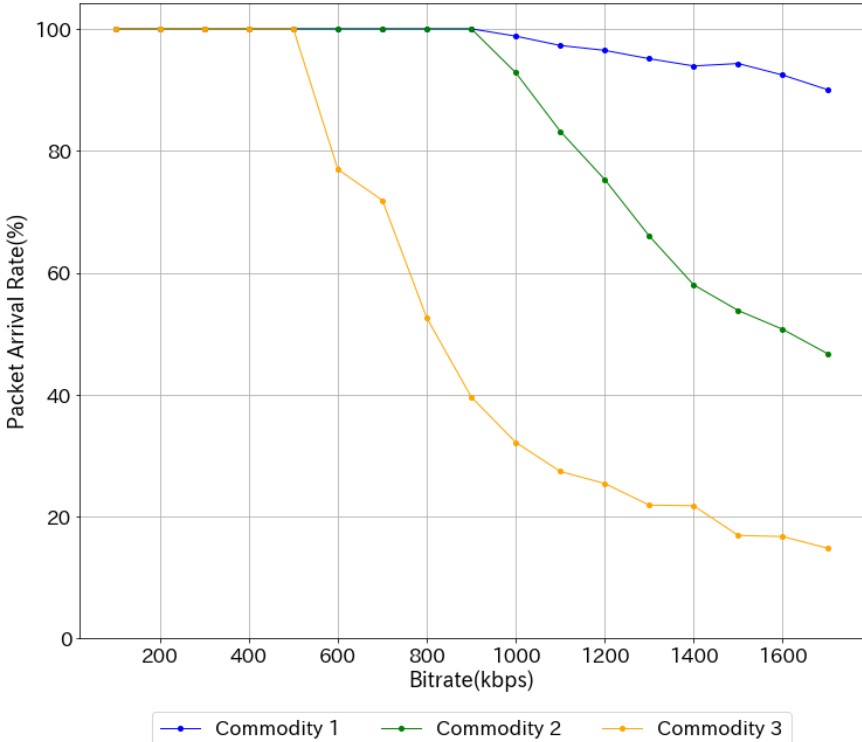

**Figure 7.** Packet arrival rate at original objective function.

The maximum flow in the graph was 82 (Figure 6), but flow bias occurred in each commodity. The packet arrival rate for commodity 1, which had the larger flow, was higher. However, that of commodity 3, which had a smaller flow, was a single path through node 11. Therefore, the arrival of commodity 3 was considerably reduced (Figure 7).

Next, Figure 8 shows the output of GLPK when using the method in Section 4.1. Figure 9 compares the results of the proposed method and OSPF in ns-3. All the objective functions of the subsequent experiments used the method of Section 4.1.

```
364 totalflow[1] B          27.3333
365 totalflow[2] B          27.3333
366 totalflow[3] B          27.3333
367 v            B          27.3333
```

**Figure 8.** Result of GLPK execution.

The maximum flow of the graph was approximately 82 (Figure 8). As the transmit bit rate increased, the packet arrival rate of the proposed method was higher than that of OSPF (Figure 9). In addition, the flow bias of each commodity was improved such that there was no longer a significant difference in the packet arrival rate of each commodity. The packet arrival rate of commodity 3 was higher than when the original objective function was used. The packet arrival rate of OSPF dropped sharply from 100%, while the proposed method showed a slower decline in arrival rate.

Table 4 compares the average end-to-end delay between OSPF and the proposed method at 200 kbps where the OSPF packet arrival rate for all commodities was 100%.

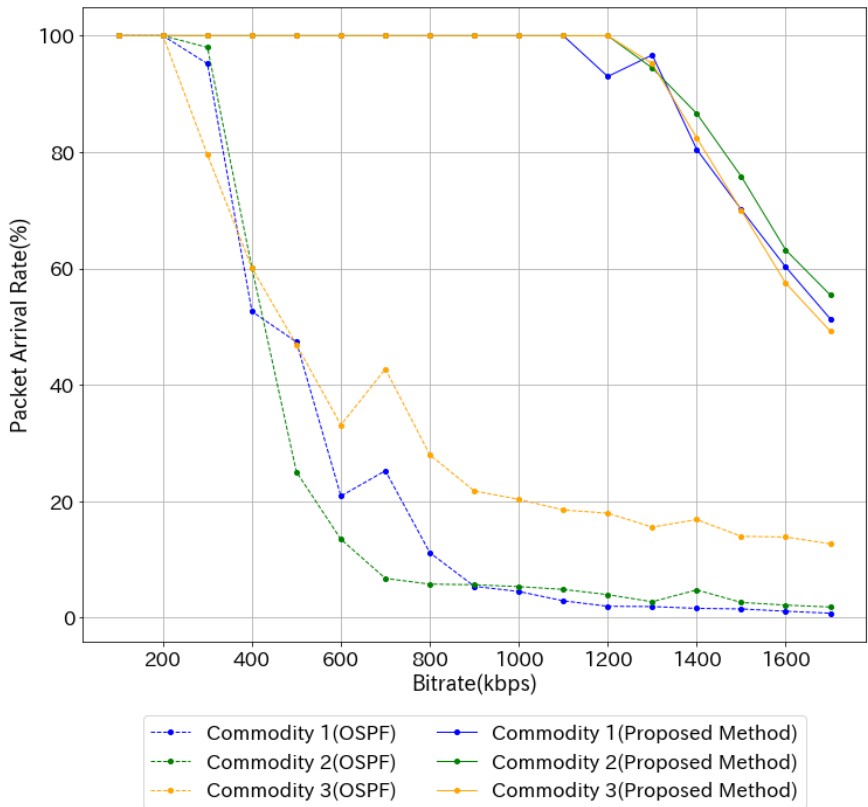

**Figure 9.** Packet arrival rate for 3 sources and 1 destination.

**Table 4.** Average end-to-end delay for 3 sources and 1 destination.

|  | OSPF | Proposed Method |
| --- | --- | --- |
| Commodity 1 | 48 ms | 54 ms |
| Commodity 2 | 31 ms | 56 ms |
| Commodity 3 | 15 ms | 22 ms |

The average end-to-end delay was shorter for OSPF than for the proposed method for all commodities (Table 4).

**Number of sources: 3, number of destinations: 3**

Figure 10 shows the result when there are three commodities in total, with commodity 1 flowing from node 1 to node 11, commodity 2 flowing from node 2 to node 10, and commodity 3 flowing from node 8 to node 7.

The proposed method maintained a higher packet arrival rate than OSPF at higher transmit bit rates, and the degree of decline in arrival rate from 100% was slower (Figure 10).

Table 5 compares the average end-to-end delay between OSPF and the proposed method at 300 kbps where the OSPF's packet arrival rate for all commodities was 100%. Only commodity 3 was experimented with at an additional 400 kbps and 500 kbps.

Basically, the average end-to-end delay was shorter for OSPF than for the proposed method (Table 5). However, at 400 kbps and 500 kbps for commodity 3, the OSPF packet arrival rate was 100%, but the value increased. Therefore, the delay in OSPF was longer than that of the proposed method.

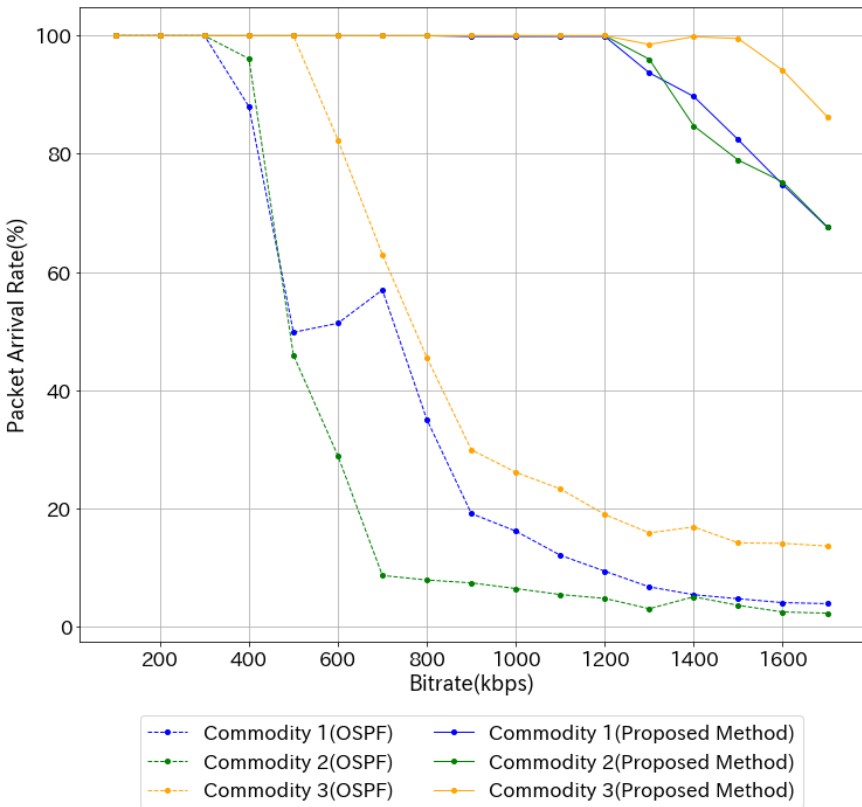

**Figure 10.** Packet arrival rate for 3 sources and 3 destinations.

**Table 5.** Average end-to-end delay for 3 sources and 3 destinations.

|  |  | OSPF | Proposed Method |
| --- | --- | --- | --- |
| Commodity 1 | 300 kbps | 59 ms | 59 ms |
| Commodity 2 | 300 kbps | 43 ms | 53 ms |
| Commodity 3 | 300 kbps | 40 ms | 46 ms |
|  | 400 kbps | 53 ms | 46 ms |
|  | 500 kbps | 421 ms | 49 ms |

### 5.2.2. Results of Experiment 2

**Number of sources: 3, number of destinations: 3**

Figure 11 shows the result when there are three commodities in total, with commodity 1 flowing from node 1 to node 11, commodity 2 flowing from node 2 to node 10, and commodity 3 flowing from node 8 to node 7. Because the bandwidth of all links is 1 Mbps, the metric for OSPF-ECMP is the number of hops.

In OSPF-ECMP, the performance of each commodity differed depending on the number of equal-cost paths. In particular, the number of paths for commodity 3 was 2, so the packet arrival rate dropped considerably. However, the proposed method achieved a higher arrival rate than any of the OSPF-ECMP commodities (Figure 11).

Table 6 compares the average end-to-end delay of OSPF-ECMP and the proposed method at 800 kbps.

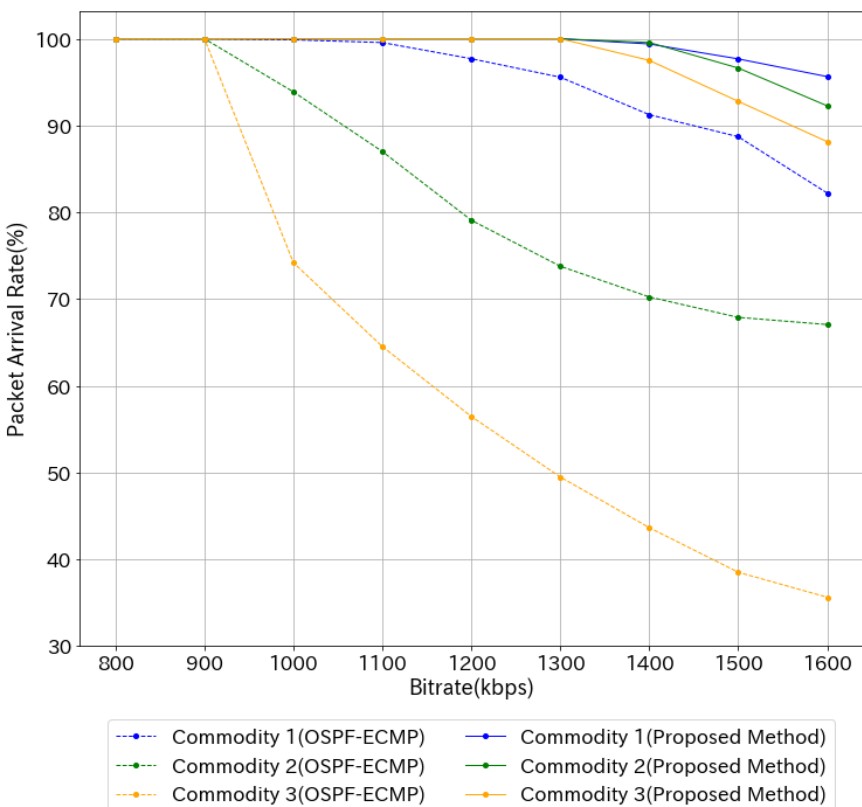

**Figure 11.** Packet arrival rate for 3 sources and 3 destinations.

**Table 6.** Average end-to-end delay for 3 sources and 3 destinations.

|  | OSPF-ECMP | Proposed Method |
| --- | --- | --- |
| Commodity 1 | 37 ms | 43 ms |
| Commodity 2 | 39 ms | 55 ms |
| Commodity 3 | 28 ms | 29 ms |

As in Experiment 1, the average end-to-end delay was shorter for OSPF-ECMP based on the optimal path than for the proposed method (Table 6).

## 6. Discussion

### 6.1. Objective Function Using the Maximin Principle

The proposed method can prevent one commodity from having zero flow and no routing. This makes it easy to apply to real networks. Adding $v$ to the objective function does not necessarily give the maximum flow of the entire network. The total flow in Figure 8 is almost equal to the maximum flow in Figure 6. Therefore, in this example, there is almost no change in network utilization even if the objective function is changed. In Figure 4, where there are only two links, the total flow is now 1 less than the maximum flow, so we can say that the network utilization has decreased. However, we think that the more links there are on the network, the more alternative paths exist that get closer to the maximum flow, so the actual network does not open up a large difference in its utilization. The maximum flow can be obtained by setting the value of the weight $w_1$ of $v$ to 0.

If flow bias occurs between commodities, the commodity with the small flow may be a single path. If the single path differs from the optimal path selected by OSPF, the packet arrival rate and average end-to-end delay for that commodity will be worse than OSPF. The proposed method satisfies the concept of Max-Min fairness and emphasizes uniform allocation. This makes it easier to prepare multiple paths for each commodity,

improving the throughput of the entire network. In particular, commodities that benefit from the proposed method are those whose flows are smaller with the original calculation method. In Figure 7, the packet arrival rate decreased considerably because the flow of commodity 3 was small and had a single path. In Figure 9, using the proposed method, the arrival rate was higher due to the increased number of paths. Conversely, commodities that can monopolize the flow with the original calculation method have smaller flows with the proposed method, which may degrade the performance of load balancing. Comparing commodity 1 in the same figure, Figure 9 has a lower reachability rate than Figure 7 because the flow is smaller due to the emphasis on fairness.

In the proposed method, by adding $v$ to the objective function, there was vertex where the objective function can be maximized rather than simply calculating the maximum flow. In this case, the values of $w_1$ and $w_2$ were set to 0.5. In examples where there is a significant difference in the bandwidth of each link, the objective function may be maximized even for small values of $v$. In that case, the value of $v$ can be increased by increasing the value of $w_1$. In general, as the value of $w_1$ is increased, the fairness among commodities increases, but the network utilization tends to decrease, as shown in Figure 4. However, there are examples where it remains almost the same, as in Figure 8.

### 6.2. Routing Using Multi-Commodity Flow Problem

The packet arrival rate of the proposed method is higher because the load is less likely to be concentrated on a particular link by providing multiple paths. The proposed method does not completely replace OSPF. Coexistence is possible because of the use of OSPF costs in capacity in the multi-commodity flow problem. OSPF is sufficient in cases where packet arrival speed is important or where the bandwidth of the shortest path is large and there is no problem if packets are concentrated on that path. In this experiment, the problem that occurs with OSPF was solved by the proposed method. Load balancing using the proposed method can be effective in networks where the path with the largest bandwidth is not too large or where there are multiple paths. Moreover, as the degree of drop in the arrival rate is slow, it is possible to prevent packets from suddenly not arriving even if the load on the network gradually increases.

OSPF-ECMP also performs load balancing, but the paths are limited to equal cost. Therefore, there is a noticeable difference in performance depending on the number of paths. When the bandwidth of a link that is the optimal path is narrow and a large amount of traffic flows there, local load balancing, such as OSPF-ECMP, may not have a significant effect, as in commodity 3 of Figure 11. The proposed method is not limited to equal cost, but considers paths in the entire network, so the number of paths is larger than in OSPF-ECMP and DFFR. The proposed method also has an advantage in that it uses the entire network without waste.

When the network is not heavily loaded, OSPF based on the optimal path has a shorter average end-to-end delay. The worst scenario of delay in the proposed method is assumed to be a situation where multiple paths are mostly composed of narrow-bandwidth links and the speed is not high. However, linear programming causes paths to include higher bandwidth links to increase the objective function. In addition, the proposed method selects paths according to a select-type group table. Therefore, links with high bandwidth are often selected, even if they contain some links with low bandwidth. In this experiment, the difference in average end-to-end delay was not significantly larger than that of OSPF. Therefore, such a scenario is not likely to occur. The proposed method using multiple paths will not have worse latency than OSPF using a single path, although latency may increase due to the flow of more traffic than expected, as in commodity 3 in OSPF in Table 5. However, from the perspective of Layer 4 of the OSI reference model, there is a worst-case scenario for delay. When the proposed method is operated with TCP, one packet may flow to another detouring path, but the performance of TCP, like DFFR, is not maintained. It is possible that the window size is exceeded and the next packet is not sent because the acknowledgement (ACK) is not returned. Therefore, we believe that it is effective

to operate the proposed method with UDP, which does not perform ordering control or retransmission control. Moreover, such problems can be alleviated by using Quick UDP Internet Connections (QUIC) [21] standardized in HTTP/3 [22]. QUIC communicates in units of streams, in which TCP-like reliability is guaranteed. This means that any delay caused by one packet using a different path will only affect the stream on which the packet resides.

Today, traffic often spikes at the start of large-scale sporting events or popular social game events. The service is not viable if packet loss occurs at a high frequency on an optimal single path. It is important from the viewpoint of service continuity to increase the packet arrival rate even if some delay occurs.

This time the path selection was on a static basis with a round-robin. We think that more flexible path control can be achieved by applying optimized multipath (OSPF-OMP) [23] to dynamically allocate paths.

## 7. Future Work

The simplex method used in this study runs in polynomial time when smoothed [24]. Many commodities flow in a real network, and it takes time to calculate a multi-commodity problem. Therefore, the use of faster algorithms and the calculation of multiple commodities in one area as one commodity are required.

In this study, the commodities were fixed and the flow table was created with proactive control. In a real network, the commodities are not fixed and the flow table needs to be updated periodically. Although it is necessary to understand all commodities in order to calculate the multi-commodity problem, it is also effective to apply the multi-commodity problem only to specific commodities, such as video distribution services.

In OpenFlow, flow entries and switch port information can be referenced by multipart requests, and the flow table can be updated by periodically sending this from the controller to the switch.

## 8. Conclusions

Network traffic has continued to increase in recent years, and existing routing protocols risk congestion due to single paths. In this study, to achieve load balancing by multiple paths, we implemented routing using the multi-commodity flow problem on OpenFlow. Compared to OSPF, it is possible to maintain a high packet arrival rate even at high transmission bit rates, demonstrating the superiority of the proposed method. Compared to OSPF-ECMP, the proposed method, which considers paths in the entire network, was more load-balancing. Although the average end-to-end delay was inferior to that of OSPF, which chooses the optimal path, the packet arrival rate is more important from the viewpoint of service continuity. Problems include the computational complexity of the simplex method used for linear programming and reactive path control, so we aim to develop a mechanism that is more suitable for the actual environment in the future.

**Author Contributions:** Conceptualization, T.F. and T.M.; methodology, T.F.; software, T.F.; validation, T.F.; formal analysis, T.F.; investigation, T.F.; data curation, T.F.; writing—original draft preparation, T.F.; writing—review and editing, T.F.; visualization, T.F.; supervision, T.M. All authors have read and agreed to the published version of the manuscript.

**Funding:** This research received no external funding.

**Data Availability Statement:** Not applicable.

**Conflicts of Interest:** The authors declare no conflict of interest.

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
