# Peer review of "Improvement of Network Flow Using Multi-Commodity Flow Problem"

_2673-8732, doi:10.3390/network3020012_

Round 1

Reviewer 1 Report

The article raises an interesting topic. Unfortunately, there is a lot of room for improvement in it.

1. The structure of the article makes it very difficult to read. There are many subsections without lead paragraphs that introduce these subsections. For example:

- "3. Preliminary Experiment" (42). "3.1. Summary" (43). 3.1 contains an introduction.

- "5.1. Objective function using the maximin principle" (108). "5.1.1. Results of experiment" (109). Section 5.1 contains only one subsection (5.1.1).

- Section 7 is very short and consists of two subsections, the first containing one paragraph and the second containing two paragraphs.

Please rewrite the article to make it more readable.

2. Lack of a good review of the literature on similar OSPF solutions (including OSPF using SDN). No new literature (except references to statistics showing network development).

Since there is no solid literature review, it is difficult to assess the novelty of the proposed solution. The idea itself is not new - as indicated, for example, by the literature cited by the authors.

Authors must add a literature review section for similar approaches (including many new publications). In this section, it should be explicitly written what the innovative solution of the authors is. For example, the authors can make a tabular summary of the existing solution and include the solution from this article in the last row of the table.

3. Section "2. Previous Research" is more of a background or introduction. Certainly not "Previous Research".

4. Lines (133-135). The tools are described too vaguely. There is no information on how OSPF, OSPF-ECMP and the proposed solution were implemented in the tested environment. Were OpenFlow switches used in all experiments?

5. The structure of section 5.2 (like many others) makes it very difficult to analyze the content (here: the analysis of the results of the experiment).

Reviewer 2 Report

First of all, I congratulate you for your work, since it shows dedication.

Many times I have had to review topics related to traffic modeling and, above all, routing, and in the vast majority of them I detect the same flaw: the number of samples and/or hops that are considered for the simulations. It is very difficult to try to argue that a mature protocol like OSPF can be displaced or have a better solution when its true nature is not considered: dealing with millions if not billions of devices since OSPF is an Interior Gateway Protocol used to distribute routing information within a single Autonomous System, and OSPF protocol was developed due to a need in the internet community to introduce a high functionality non-proprietary Internal Gateway Protocol (IGP) for the TCP/IP protocol family. Very interesting work from the mathematical point of view at the laboratory level, it does not translate into a real proposal that could eventually result in the replacement of OSPF.

The bibliography is completely outdated, the most current reference corresponds to the year 2020, the rest are as old as number 6, for example, which is from 1962. It must be treated again, it is essential.

The experiments, although they are adequately formulated and developed, do not really account for how the proposal can improve the operation of OSPF, nor does it indicate any real comparative way with OSPF, but nevertheless as a work to be developed in a laboratory for a computer network subject, it's perfect. In fact, you yourselves acknowledge that your simulations are far from realistic in the "future work" section. All in all, then how does your proposal outperform OSPF in a real world scenario? There then remains a reasonable doubt regarding the impact of the article beyond the academic sphere.

Reviewer 3 Report

- Problem description

Due to the widespread use of the Internet, the volume of its traffic has been increasing.  However, existing routing protocols that select only one optimal path are insufficient to control congestion in such scenarios. Such protocols tend to overload certain links, thereby increasing the risk of congestion. To overcome this limitation, this study proposes a multipath control mechanism using a multi-commodity flow problem. By comparing the proposed method with single-path control (OSPF) and multipath control (OSPF-ECMP), authors confirmed that the proposed method improves the packet arrival rate, which is expected to alleviate congestion. Furthermore, since the network contains redundant paths to the destination node, this approach exploits such paths to balance the load on the network links.

- Main insights in the paper

Existing methods cannot maximize the network utilization of data path.

  1. The Open Shortest Path First (OSPF) algorithm is designed to minimize the cost and latency of each flow, but it lacks the capability to maximize network and path utilization. This means that even though it can select the shortest path for a data flow, it may not utilize the network to its full potential.
  2. OSPF-ECMP (Equal Cost Multipath) is a load balancing technique that selects multiple paths with equal cost and latency to distribute data flows. However, this technique is limited to situations where multiple paths have the same cost and latency, which means that it may not be effective in scenarios where there are multiple paths with different costs or latencies.

- Core contributions of the paper

  1. In order to achieve maximum network utilization, the authors of the study have mapped the problem to the multi-commodity flow problem. By doing so, they are able to better understand the complex interactions between different data flows and how they can be optimized to maximize the network's capacity and performance.
  2. To address the challenge of load balancing across multiple paths, the authors have introduced a new objective function designed to ensure that each commodity has at least some active flows. This means that the network is able to efficiently distribute traffic across multiple paths, reducing congestion and ensuring that each data flow is able to reach its destination in a timely and efficient manner.

- detailed comments on core technical sections in the paper

  1. I have a question regarding the effectiveness of the new objective function introduced in the study. While it aims to ensure fairness among multiple commodities, it is unclear to me how it leads to better load balancing among paths. I would greatly appreciate it if the author could provide more clarification on this topic.
  2. I have a concern about determining the appropriate values for w1, w2, and v in practice. Additionally, with the introduction of the new objective function, it is unclear to me how close the actual network utilization can come to the optimal network utilization. Could the author kindly provide more insights or guidance on this topic?
  3. I am curious about the implementation details discussed in section 4.2.3, specifically regarding the use of OpenFlow for multi-path routing. I was wondering whether this is performed on a per-packet or per-flow basis. If it is done on a per-packet basis, I am concerned about the potential degradation of TCP performance due to packet reordering. Alternatively, if it is done on a per-flow basis, I am wondering whether it is assumed that each flow has an equal rate. If not, there may be suboptimal performance as an elephant flow could select the wrong path due to round-robin selection. I would appreciate it if the authors could provide further clarification on this topic.

- detailed comments on the evaluation section in the paper

  1. The author of the study has compared the proposed solution against OSPF and OSPF-ECMP, and has demonstrated superior network utilization. However, I would appreciate more information on the performance difference between the proposed scheme and the original solution solved by the multi-commodity flow problem. This would help me better understand the relative benefits of the proposed approach.
  2. To better understand the trade-off between fairness and network utilization, I believe it would be valuable for the author to provide a sensitivity test for w1 and w2. This would help readers gain a deeper understanding of the potential impact of these parameters on the overall performance of the network.
  3. I am curious about the worst-case scenario for the proposed scheme in terms of latency, and how it compares to OSPF. Additionally, I would appreciate it if the author could provide insights on any potential solutions to help reduce latency in such scenarios. 

- overall summary of the review

  1. The authors have demonstrated that by mapping the routing problem to a multi-commodity flow problem, their proposed solution achieves higher network utilization than existing approaches such as OSPF and OSPF-ECMP. 
  2. I have a question regarding the relationship between ensuring fairness among multiple commodities and load balancing among multi-paths. I would appreciate it if the authors could provide more clarification on how these two concepts are related and how they impact network utilization. Additionally, I am curious about whether maximizing network utilization implies that the paths are well-utilized, and whether ensuring fairness should be treated as a separate objective. 
  3. While the traffic engineering problem is well-known and long-lasting, I was surprised to see that the author did not include any related works that may have attempted to achieve similar goals. I believe it would be beneficial for the author to discuss related works in the discussion section, as this would provide readers with a more comprehensive understanding of the research landscape.

Round 2

Reviewer 1 Report

Thank you for considering the reviewer's comments. Nevertheless, some comments from the previous review have still not been sufficiently addressed.

The tools are described too vaguely.

There is no information on how OSPF, OSPF-ECMP and the proposed solution were implemented in the tested environment.

Reviewer 3 Report

I checked out the revision authors did and it looks good! Authors put in a lot of effort and attention to detail, and it really shows in the current draft.

Author Response

Thank you very much for providing important comments.